# Volumetric Assessment of the Frontal Sinus in Female Adolescents and Its Relationship with Craniofacial Morphology and Orthodontic Treatment: A Pilot Study

**DOI:** 10.3390/ijerph19127287

**Published:** 2022-06-14

**Authors:** Masaki Sawada, Hiroshi Yamada, Masaaki Higashino, Susumu Abe, Eiji Tanaka

**Affiliations:** 1Yamada Orthodontic Office, Izumiotsu 595-0025, Japan; sawada@88-ortho.com (M.S.); yamada@88-ortho.com (H.Y.); 2Department of Otorhinolaryngology, Head and Neck Surgery, Osaka Medical and Pharmaceutical University, Takatsuki 569-8686, Japan; masaaki.higashino@ompu.ac.jp; 3Department of Comprehensive Dentistry, Tokushima University Graduate School of Biomedical Sciences, Tokushima 770-8504, Japan; susumu.abe@tokushima-u.ac.jp; 4Department of Orthodontics and Dentofacial Orthopedics, Tokushima University Graduate School of Biomedical Sciences, Tokushima 770-8504, Japan

**Keywords:** computed tomography, craniofacial morphology, frontal sinus, paranasal sinus

## Abstract

The present study aimed to evaluate the correlation between frontal sinus morphology and craniofacial morphology, and to investigate the effects of orthodontic treatment on the development of the frontal sinus in female adolescents (mean age: 13.9 ± 1.3 years). In total, 53 patients were recruited and underwent cephalography and computed tomography before and after orthodontic treatment. Of these patients, most had a bilaterally symmetrical fan-shaped frontal sinus without any fusion. The average size and volume of the frontal sinus before orthodontic treatment were 45.8 ± 12.3 mm in breadth, 29.8 ± 7.3 mm in height, 22.7 ± 5.1 mm in depth, and 5151.6 ± 2711.4 mm^2^ in volume. Sinus volume in patients with skeletal Class III malocclusion tended to be larger than that in patients with skeletal Class I and II malocclusion. Upon comparison with the pretreatment measurements, the sizes and volumes of the frontal sinus were significantly larger following orthodontic treatment, regardless of the skeletal pattern; however, since these changes were small, the increases in the size and volume of the frontal sinus may have been caused by pubertal growth and not orthodontic treatment. The clinical relevance of the frontal sinus remains controversial.

## 1. Introduction

The paranasal sinuses, the hollow spaces in the craniofacial bones around the nose, are air-filled cavities within the frontal, ethmoidal, sphenoidal, and maxillary bones [1]. All the sinuses drain into the superior and lateral aspects of the nose [2], and the lining mucosa of the sinuses connects to the nasal cavity. Paranasal sinuses are complex anatomical structures, characterized by highly variable shapes, morphologies, and sizes.

The paranasal sinuses occupy a significant amount of space in the cranium and have been of interest in studies that seek to determine their function and the factors affecting their structure and size [3]. The paranasal sinuses have essential functions in immune defense and air filtration processes carried out by the nose, and the walls of the sinus cavities are lightly coated with mucus, which keeps the tissue moist and healthy and also traps bacteria. Furthermore, Preuschoft et al. [4] reported that the paranasal sinuses develop in response to the biomechanical requirements of the skull architecture. Thus, the magnitude and direction of the mastication forces, which are major contributors in mechanical stress induction, may be of great importance for paranasal sinus development.

The frontal sinus is the most complex of the paranasal sinuses owing to its location, anatomical variations, and multiple clinical presentations. The frontal sinuses are absent at birth and are generally well-developed in childhood. They continue to grow gradually until they reach their maximum size after puberty. Amusa et al. [5] examined 24 dried skulls of a Nigerian population and reported 58% frontal sinus aplasia, which implies that the frontal sinus might not be a vital organ in the living body. Nevertheless, it is important to pay great attention to the role and function of the frontal sinus, since changes in the size of the frontal sinus could be used as an indicator of harmonious anterior occlusion [3].

Our study was designed to determine the normal size and volume of the frontal sinus in female adolescents using three-dimensional (3D) computed tomography (CT), and evaluate the correlation between craniofacial morphology and frontal sinus morphology. Furthermore, we investigated the effects of orthodontic treatment on frontal sinus development. This information may have clinical implications for the prognosis of orthodontic treatment of various malocclusions with craniofacial discrepancies.

## 2. Materials and Methods

### 2.1. Participants

Participants who underwent conventional orthodontic treatment at the Yamada Orthodontic Office between January 2010 and December 2019 were recruited for this study. Informed consent was obtained from all the participants after a full explanation of the research purposes and procedures. The exclusion criteria were as follows: history of seasonal allergies; ear, nose, and throat-related diseases; hormonal disturbances; any deformity or disease in the craniofacial region; and previous orthodontic treatment. This study was approved by the Ethics Committee of Tokushima University Hospital (approval no. 3900).

The present study estimated the necessary sample size to meet the desired statistical constraints. The effect size was used for convenient statistical parametric and non-parametric tests. The effect size of the comparison among the three subgroups divided according to the maxillomandibular jaw–base relationship was considered medium (0.25). The statistical power (1-β) was calculated using G*Power software. Power analysis was based on one-way or two-way repeated measurement analysis of variance (r-ANOVA) with a medium effect size of 0.25, significance level (Type I error) of 0.05, and power level of 0.8. Power analysis was performed and the total estimated sample size was determined to be 42 when a two-way r-ANOVA was used.

For all the patients, 3D CT was performed before orthodontic treatment using a CT system (Alphard-3030, Asahi Roentgen Ind. Co., Ltd., Kyoto, Japan) with the following acquisition parameters: 60–110 kV; 3–15 mA; collimation, 0.6 mm; rotation time, 18 s; and reconstruction thickness, 0.39 mm. Imaging data were processed using Dolphin Imaging (Dolphin Imaging & Management Solutions, Verona, Italy) for orthodontic diagnosis and treatment planning. Using a series of CT DICOM data, a 3D model of the frontal sinus and volume-rendered images were extracted. CT was also performed after active orthodontic treatment. Furthermore, lateral cephalograms were obtained before and after orthodontic treatment using a cephalometric radiographic system (Hyper-X CM, Asahi Roentgen Ind. Co., Ltd., Kyoto, Japan). All cephalograms were obtained with the teeth in the intercuspal position. Briefly, the participant’s head was fixed with ear rods and stabilized in a position such that the Frankfort horizontal plane was parallel to the floor.

Each lateral cephalogram was traced on acetate paper by one examiner (H.Y.). The tracings were computerized using a graphic digitizer (Dolphin Imaging, Dolphin Imaging & Management Solutions, Verona, Italy) by another examiner (M.S.) to obtain measurements of the craniofacial morphology. The accuracy of the tracing was confirmed by two orthodontic experts who joined this study as collaborators. All the investigators were blinded to the participants’ general status. Before taking the measurements, the intra-examiner reliability of cephalometric analysis was determined using the intraclass correlation coefficient (ICC) on 20 randomly selected cephalograms that were traced and plotted with three arbitrary points (nasion, sella, and pogonion points) by the same examiner twice within an interval of one week. As a result, the ICC was 0.984, confirming a sufficient reliability of the selected measurements.

### 2.2. Craniofacial Morphology

Based on the cephalometric measurements, the participants were divided into the following three subgroups according to the ANB angle (angle between the nasion-A-point and nasion-B-point lines); skeletal Class III group (participants with ANB angle < 1.0°), skeletal Class I group (ANB angle ≥ 1.0° but < 5.0°), and skeletal Class II group (ANB angle > 5.0°). From the lateral cephalogram, 13 angular and 10 linear measurements were analyzed for morphometric evaluation (Table 1).

### 2.3. Frontal Sinus Morphology

The 3D models constructed from CT Digital Imaging and Communications in Medicine (DICOM) data were analyzed using three-dimensional viewing software to automatically measure the maximum breadth, height, and depth of the frontal sinus (Figure 1). The sinus volume was determined as an integral volume of the air cavity within the bony walls of the sinus in the frontal bone on the reformatted axial, sagittal, and coronal images. Volume-rendering images were used for the automatic calculation of the frontal sinus volume. The maximum width was measured between the most lateral points of the frontal sinus. The maximum height was measured between the baseline and the highest point of the frontal sinus. The maximum breadth was defined as the length between the most prominent point of the anterior and posterior parts of the frontal sinus. Furthermore, using a 3D frontal sinus model, morphological variations, including bilateral or unilateral, symmetrical or asymmetrical, fusion or separation, and presence or absence of changes in shape, were analyzed. According to the classification of previous studies [6,7], the shape of the frontal sinus in an anterior view was divided into three types: fan-shaped, quadrangular, and irregular. The term “separation” was defined if the frontal sinus was divided into two or more segments, while the term “fusion” denoted the frontal sinus connecting with two or more segments.

### 2.4. Statistical Analysis

Statistical analyses were performed using SPSS 27.0 (SPSS Inc., Chicago, IL, USA). The normality of each morphometric variable was assessed using the Shapiro–Wilk test. The average size and volume of the frontal sinus before orthodontic treatment were calculated for each subgroup according to the ANB angle. In addition, the differences in the sinus size and volume between the pretreatment and posttreatment stages were evaluated and compared among the three subgroups. Morphological features of the frontal sinus were verified by subgroup analysis using the Fisher’s exact test. For data with normal distribution, a general linear model analysis for repeated measures was performed to compare the three subgroups. Intergroup comparisons were performed using the paired *t*-test with the Bonferroni method as a post-hoc test. A linear single regression test was performed to detect the relationships between the morphological variables of the frontal sinus and the cephalometric measurement variable for each subgroup. Moreover, multiple regression analysis, including the morphological variables regardless of the subgroups, was performed to assess the relationships of the morphological variables as a response variable and the cephalometric measurement variables as explanatory variables. Probabilities below 0.05 as type I error (α) were considered statistically significant.

## 3. Results

### 3.1. Participants

The total sample size used in the present study was 53. The participants were divided into three subgroups: skeletal Class I, 20 females ranging from 11.9 to 17.3 years (mean age ± standard deviation [SD]: 13.9 ± 1.3 years); skeletal Class II, 20 females ranging from 11.7 to 16.4 years (mean age ± SD: 13.9 ± 1.4 years); and skeletal Class III, 13 females ranging from 10.3 to 15.7 years (mean age ± SD: 13.4 ± 1.6 years). There was no significant difference in age among the three subgroups (*p* = 0.53, one-way ANOVA). The duration of orthodontic treatment was 3.8 ± 1.1 years.

### 3.2. Volumetric and Geometric Measurements of the Frontal Sinus and Its Relation with the Skeletal Pattern

Considering the morphological features of the frontal sinus, no patients exhibited agenesis of the frontal sinus. Of the 53 patients, most of the patients (98.1%) had a bilateral frontal sinus without any fusion (Table 2). Considering the shape of the frontal sinus, 42 patients (79.2%) exhibited symmetry, while 11 patients (20.8%) showed asymmetry. There were no statistically significant differences among the three subgroups (*p* = 0.43). Furthermore, 38 patients (71.7%) showed a fan-shaped sinus, followed by irregular (18.9%), and quadrangular sinuses (9.4%). The morphological shapes did not differ significantly among the three subgroups (*p* = 0.92). The morphological characteristics of the frontal sinus were not associated with the maxillomandibular jaw–base relationship (skeletal pattern), indicating no specific features of the frontal sinus (*p* = 0.37).

Before orthodontic treatment, the breadth, height, and depth of the frontal sinus were 46.2 ± 12.5 mm (Mean ± SD), 29.8 ± 8.0 mm, and 22.8 ± 5.1 mm in the skeletal Class I; 44.8 ± 10.5 mm, 31.0 ± 6.2 mm, and 22.6 ± 4.9 mm in the skeletal Class II; and 46.5 ± 15.7 mm, 27.8 ± 8.2 mm, and 22.7 ± 6.2 mm in the skeletal Class III malocclusion, respectively (Table 3). No significant differences in the size of the frontal sinus were observed among the three subgroups. The volume of the frontal sinus was 4986.3 ± 2849.2 mm^2^, 5143.3 ± 2398.9 mm^2^, and 5418.9 ± 3221.9 mm^2^ in the skeletal Class I, II, and III groups, respectively. No significant difference in the sinus volume was observed among the three subgroups (*p* > 0.63); however, the sinus volume in the skeletal Class III group tended to be larger than that in the remaining two subgroups.

By comparing pretreatment and posttreatment measurements, considerably large changes in the size and volume of the frontal sinus were observed following orthodontic treatment, regardless of the skeletal pattern (Table 3). Although there was no significant interaction among the three subgroups, the breadth, height, depth, and volume of the frontal sinus significantly (*p* < 0.001) increased during orthodontic treatment, regardless of the skeletal classification.

### 3.3. Correlation between the Craniofacial Morphology and Frontal Sinus Morphology

The standardized coefficients were calculated for four measurement values of the frontal sinus and 23 cephalometric variables using single regression analysis. Among the 92 correlations for all the participants, most coefficients had very weak or no correlations without statistical significance between pretreatment and posttreatment measurements (Table 4a). For the participants with skeletal Class I malocclusion, 2 of 92 correlations showed weak but significant (*p* < 0.05) correlations with sinus volume and FMA in the pretreatment measurements, and with sinus height and IMPA in the posttreatment measurements (Table 4b). For the participants with skeletal Class II, 4 of 92 correlations had significantly (*p* < 0.05, *p* < 0.01) negative correlations with sinus breadth and volume and palatal plane to FH, sinus width and L1-NB, and sinus height and overbite, and a significant positive correlation with sinus width and FMIA in the posttreatment measurements; however, no significant correlations were found for the pretreatment measurement variables (Table 4c). For the participants with skeletal Class III malocclusion, 11 of 92 correlations revealed weak or mild but significant (*p* < 0.05, *p* < 0.01) correlations with sinus breadth, width, and volume and FMA; sinus breadth, width, and volume and IMPA; sinus width, height, and volume and gonial angle; sinus width and Y-axis; and sinus width and facial angle at the pretreatment measurements (Table 4d). Furthermore, in the posttreatment measurements, significant (*p* < 0.05, *p* < 0.01) positive correlations were found between sinus breadth, width, height, and volume and Y-axis; sinus breadth, width, height, and volume and FMA; and sinus volume and gonial angle. Three correlations had significant (*p* < 0.05) negative correlations with sinus breadth, sinus volume, and IMPA; and sinus width and facial angle. Multiple regression analysis was applied to verify the relationships of the frontal sinus size and volume with the horizontal skeletal pattern as a qualitative variable, and the cephalometric measurement variable as a quantitative variable, in this study. However, no significant relationship was observed between the skeletal pattern and sinus size and volume. Thus, the difference in the skeletal pattern did not affect the sinus size and volume.

## 4. Discussion

With technological advancements in CT, growing evidence suggests the critical role of paranasal sinuses in craniofacial growth and orthodontic treatment [7,8,9,10,11,12,13,14,15,16]. In the present study, the three-dimensional size and volume of the frontal sinus were measured using CT images taken before orthodontic treatment in female adolescents. The average size and volume of the frontal sinus before orthodontic treatment were 45.8 ± 12.3 mm in breadth, 29.8 ± 7.3 mm in height, 22.7 ± 5.1 mm in depth, and 5151.6 ± 2711.4 mm^2^ in volume. These values were nearly consistent with those of previous studies [8,9,17]. Furthermore, no age-related differences in the frontal sinus size and volume were found in the present study, similar to previous studies [8,9]. Using radiographic examination of the frontal sinus, Brown et al. [18] reported that the main expansion of the sinus ceased at the age of 15.68 years in males and 13.72 years in females. On the other hand, growth ceases at approximately 20 years of age, when the shape and size of the frontal sinus become stable [10]. Considering the mean age of our patients was 13.8 ± 1.4 years before orthodontic treatment, the size and volume of the frontal sinus were assumed to already be at their largest.

Meanwhile, our results showed that the three-dimensional sizes and volume of the frontal sinus increased significantly during orthodontic treatment, while the morphological features of the sinus were not completely changed. As described above, since the mean age of our patients was 13.8 ± 1.4 years before orthodontic treatment, no or minimal changes in the frontal sinus size were found if they had not received orthodontic treatment. Kjær et al. [19] reported the importance of environmental factors on frontal sinus dimensions, indicating that medical treatment, including orthodontic and orthognathic treatment, may affect the size of the frontal sinus. This suggests that orthodontic treatment may contribute to the development of the frontal sinus as an environmental factor, even after pubertal growth of the frontal sinus is completed. However, since the increases in the size and volume of the frontal sinus were small in this study, these may have been caused by pubertal growth and not orthodontic treatment. Further studies involving control participants without an experience of orthodontic treatment are needed to examine whether orthodontic treatment affects the dimensions of the frontal sinus.

According to Rae and Koppe [20], possible functions of the paranasal sinuses include respiratory function, thermoregulation, and trauma protection to decrease the skull weight. Preuschoft et al. [4] reported that the paranasal sinuses develop in response to the biomechanical requirements of the skull architecture. Thus, the magnitude and the direction of the forces of mastication, which are major contributors to mechanical stress induction, are of great importance. Furthermore, Said et al. [3] investigated the relationship between anterior occlusion and frontal sinus size in adolescents using cephalograms, and reported that the frontal sinus size could be used as an indicator for harmonious anterior occlusion. This implies that a larger frontal sinus has favorable functions, such as serving as a shock absorber and in the transmission of occlusal forces. On the other hand, Benington et al. [21] demonstrated that the group with the largest sinus size was the open bite group, which might be attributed to the reduced transmission of occlusal forces along the nasal pillars because of the lack of contact between the maxillary and mandibular incisors, and weaker muscles associated with the hyperdivergent morphology. Furthermore, recently, Celiker et al. [11] investigated the relationship between the size of the frontal sinus and mortality in patients with cranial trauma, and suggested that the larger the sinus, the greater the risk of death resulting from trauma to the head. Our results showed very weak correlation between the three-dimensional size and volume of the frontal sinus and overbite, except a significant, but weak, negative correlation between the sinus height and overbite after the treatment in patients with skeletal Class II malocclusion. Based on the previous studies, a frontal sinus of proper size may have favorable functions for anterior occlusion in adolescents; however, an extraordinarily large sinus could contribute to one of the risk factors for death following severe head injury. Therefore, the decisive role of the frontal sinus remains controversial.

Previously, the association between frontal sinus size and volume and craniofacial morphology was evaluated using lateral cephalograms. Rossouw et al. [22] found a correlation between the frontal sinus area on lateral cephalometric radiographs and maxillary length, mandibular length, symphysis width, and condylar length, indicating that the frontal sinus size may be a supplementary indicator for mandibular growth prediction. Benington et al. [21] also demonstrated that anterior cranial base length, facial divergence, and inclination of the maxillary incisor in relation to the palate were statistically significant variables explaining frontal sinus size. Recently, Tehranchi et al. [23] demonstrated that a larger frontal sinus size was associated with reduced inclination of the anterior cranial base, increased anterior facial height in males, and increased gonial angle in females. Furthermore, Yassaei et al. [24] indicated that the dimensions and surface area of the frontal and maxillary sinuses in skeletal Class III malocclusion were greater than those in other groups. These variables (except for frontal sinus width) were significantly correlated with the anterior and posterior cranial bases and mandibular body length. In the present study, we evaluated the correlation between craniofacial morphology and frontal sinus morphology in Japanese female adolescents using 3D CT. No significant correlation was found among the 92 correlations for all the participants. For each group with different skeletal patterns, several cephalometric variables demonstrated weak or mild, but significant correlations with the frontal sinus size and volume. For the participants with a skeletal Class III jaw–base relationship, the values of the Y-axis and FMA were significantly positively correlated with the breadth, width, height, and volume of the frontal sinus, indicating that the longer the facial type, the greater the height of the frontal sinus. Furthermore, our results indicated a frontal sinus that was larger, but not significant, in patients with skeletal Class III malocclusion than in those with skeletal Class I and II malocclusions, which was consistent with the previous results [24,25]. However, there was no significant difference in frontal sinus size among the participants with the same skeletal classification and different anterior occlusions. Further studies are needed to categorize treatment modalities and to perform more detailed statistical investigations. In addition, a larger number of participants should be included in future studies.

This study had some limitations. First, the study did not include any control samples, that is, participants who did not undergo orthodontic treatment. Our results showed small, but significant, increases in the size and volume of the frontal sinus during orthodontic treatment; however, without the control data, we could not distinguish whether these increases were affected by orthodontic treatment. It is difficult to record 3D CT images for individuals who are not undergoing treatment because of ethical concerns. Second, our participants were only females, and we could not estimate the sex-dependent differences in the dimensions of the frontal sinus. Furthermore, the effect of orthodontic treatment on the sinus dimensions may differ between male and female adolescents. Previous studies also did not report a sex-based difference in the size and volume of the frontal sinus [3,10]; therefore, further studies involving both male and female participants are needed.

## 5. Conclusions

In conclusion, the present study reported the average size and volume of the frontal sinuses of female adolescents before orthodontic treatment. During orthodontic treatment, the sizes and volume of the frontal sinus increased significantly regardless of the skeletal classification; however, since these changes were small, the increases in the size and volume of the frontal sinus may have been caused by pubertal growth and not orthodontic treatment. Further studies should be conducted to examine the causative effect of orthodontic treatment on the development of the frontal sinus.

## Figures and Tables

**Figure 1 ijerph-19-07287-f001:**
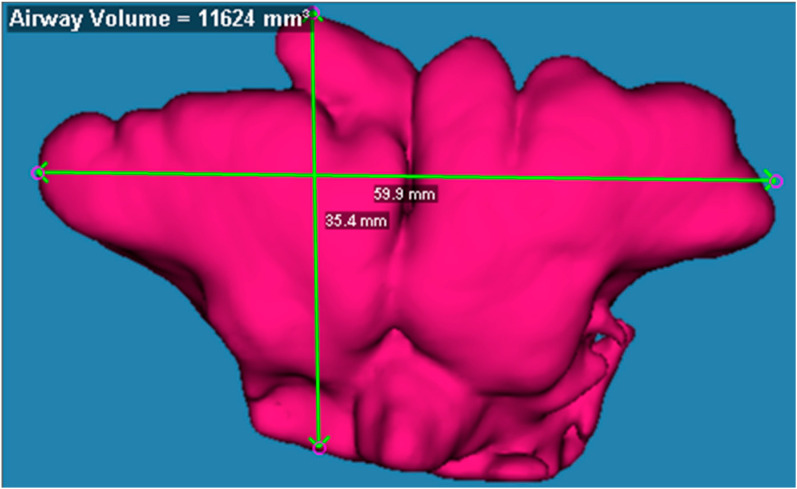
A representative image of the frontal sinus.

**Table 1 ijerph-19-07287-t001:** Definitions of cephalometric measurement items.

** *Angular Measurement Items (°)* **
**SNA**: Angle between the sella–nasion and nasion-A-point lines, indicating the anteroposterior position of the maxilla relative to the anterior cranial base.
**SNB**: Angle between the sella–nasion and nasion–B-point lines, indicating the anteroposterior position of the mandible relative to the anterior cranial base.
**ANB**: Angle between the nasion-A-point and nasion–B-point lines, indicating an anteroposterior relationship between the maxilla and mandible.
**Facial angle**: Angle between the nasion–pognion line and Frankfort horizontal plane, indicating chin prominence.
**Y-axis**: Angle between the sella–gnathion line and Frankfort horizontal plane, indicating the mandibular growth direction.
**Gonial angle**: Angle between the mandibular and ramal planes.
**FMA**: Angle between the mandibular plane and Frankfort horizontal plane, indicating divergence of the mandibular plane.
**Occlusal plane to SN**: Angle between the occlusal plane and sella–nasion line, indicating the inclination of the occlusal plane.
**Palatal plane to FH**: Angle between the palatal plane and Frankfort horizontal plane, indicating inclination of the palatal plane.
**U1 to SN**: Angle between the long axis of the maxillary central incisor and sella–nasion line, indicating labiolingual inclination of the upper incisor.
**Interincisal angle**: Angle between the long axes of the upper and lower incisors.
**IMPA**: Angle between the long axis of the mandibular incisor and the mandibular plane, indicating the labiolingual inclination of the mandibular central incisor relative to the mandibular plane.
**FMIA**: Angle between the long axis of the mandibular central incisor and Frankfort horizontal plane, indicating the labiolingual inclination of the mandibular incisor relative to the Frankfort horizontal plane.
** *Linear Measurement Items (mm)* **
**SN**: Distance between the sella and the nasion, indicating the anteroposterior length of the anterior cranial base.
**U1 to NA**: Distance between the incisal edge of the maxillary central incisor and the line joining the A-point to nasion.
**L1 to NB**: Distance between the incisal edge of the mandibular central incisor and the line joining the B-point to the nasion.
**Overjet**: Anteroposterior distance between the maxillary and mandibular central incisal edges.
**Overbite**: Vertical dimension between the maxillary and mandibular central incisal edges.
**Wits appraisal**: Perpendicular distance between points A and B on the occlusal plane, indicating the degree of anteroposterior jaw disharmony.
**N–Me**: Distance between nasion and menton, indicating anterior facial height.
**Ar–Go**: Distance between the articulare and gonion, indicating the length of the mandibular ramus.
**Ar–Me**: Distance between the articulare and menton, indicating the effective mandibular length.
**Go–Me**: Distance between the gonion and menton, indicating the length of the mandibular corpus.

**Table 2 ijerph-19-07287-t002:** Morphological features of the frontal sinus.

	Skeletal Class I	Skeletal Class II	Skeletal Class III	Total
Bilateral or unilateral				
Bilateral	20	20	2	52 (98.1%)
Unilateral	0	0	1	1 (1.9%)
Symmetry or asymmetry				
Symmetry	17	14	11	42 (83.0%)
Asymmetry	3	6	2	11 (17.0%)
Spatial relationship				
Fusion	0	0	0	0 (0%)
Separation	20	20	13	53 (100%)
Morphological shape				
Fan-shaped	13	15	10	38 (71.7%)
Irregular	5	3	2	10 (18.9%)
Quadrangular	2	2	1	5 (9.4%)
Posttreatment morphology				
No changes	18	20	12	50 (94.3%)
Change	2	0	1	3 (5.7%)

**Table 3 ijerph-19-07287-t003:** The frontal sinus size and volume for each subgroup before and after orthodontic treatment.

		Orthodontic Treatment		*p*-Value		
		Before	After	Interaction	Class	Time
Breadth (mm)	Class I	46.2 ± 12.5	48.3 ± 12.2	0.179	0.843	**<0.001**
	Class II	44.8 ± 10.5	46.1 ± 10.4			
	Class III	46.5 ± 15.7	49.4 ± 17.2			
	All	45.8 ± 12.3	47.7 ± 12.7			
Height (mm)	Class I	29.8 ± 8.0	31.6 ± 7.9	0.6642	0.546	**<0.001**
	Class II	31.0 ± 6.2 *	32.4 ± 6.4 *			
	Class III	27.8 ± 8.2	29.8 ± 8.9			
	All	29.8 ± 7.3	31.5 ± 7.5			
Depth (mm)	Class I	22.8 ± 5.1	24.8 ± 6.5	0.433	0.92	**<0.001**
	Class II	22.6 ± 4.9	23.6 ± 5.2			
	Class III	22.7 ± 6.2	23.9 ± 6.9			
	All	22.7 ± 5.1	24.1 ± 6.0			
Volume (mm^2^)	Class I	4986.3 ± 2849.2	5644.8 ± 2939.5	0.626	0.896	**<0.001**
	Class II	5143.3 ± 2398.9	5620.7 ± 2536.6			
	Class III	5418.9 ± 3221.9	6129.0 ± 3698.3			
	All	5151.6 ± 2711.4	5754.4 ± 2923.5			

*: This value was not mathematically normally distributed in the results of the normality test. However, since an evaluation of the Q–Q plot or the graph using data distribution were close to the normal distribution, the statistical analysis was performed as for normal distribution. General linear model analysis for repeated measures was performed with class (Class I, Class II, and Class III), time (before and after orthodontic treatment), and their interaction. Bold text shows variables with significant difference between the conditions.

**Table 4 ijerph-19-07287-t004:** Standardized coefficients between the measurement values for the frontal sinus in relation to the cephalometric measurement variables analyzed by single regression analysis. (**a**) All participants, (**b**) participants with skeletal Class I jaw–base relationship, (**c**) participants with skeletal Class II jaw–base relationship, and (**d**) participants with skeletal Class III jaw–base relationship.

**(a) All participants**																							
Pretreatment measurements																							
	Angular measurements													Linear measurements									
	SNA	SNB	ANB	Facial	Y-axis	Occ.pl.	Go.A	FMA	Pal pl.	U1-SN	IIA	IMPA	FMIA	SN	U1-NA	L1-NB	OJ	OB	Wits	N-Me	Ar-Go	Ar-Me	Go-Me
Frontal sinus																							
breadth	−0.200	−0.146	−0.066	−0.130	0.103	0.073	0.119	0.061	0.088	−0.236	0.2	−0.114	0.068	−0.033	−0.074	−0.080	−0.127	−0.092	−0.048	−0.026	0.077	−0.100	−0.110
width	−0.199	−0.287	0.121	−0.233	0.181	−0.177	−0.104	−0.028	0.015	−0.236	0.057	0.15	−0.129	0.054	−0.019	0.091	0.051	−0.043	0.022	0.035	0.142	−0.220	−0.194
height	−0.114	−0.080	−0.042	−0.103	0.113	−0.037	0.059	0.075	0.075	−0.058	0.112	−0.133	0.077	0.006	0.04	−0.050	0.022	−0.068	0.024	0.036	0.026	−0.019	−0.040
volume	−0.167	−0.123	−0.055	−0.156	−0.106	0.032	0.065	0.008	0.109	−0.177	0.146	−0.028	0.021	0.028	−0.017	−0.031	−0.104	−0.048	−0.003	−0.052	0.106	−0.091	−0.143
Posttreatment measurements																							
	Angular measurements													Linear measurements									
	SNA	SNB	ANB	Facial	Y-axis	Occ.pl.	Go.A	FMA	Pal pl.	U1-SN	IIA	IMPA	FMIA	SN	U1-NA	L1-NB	OJ	OB	Wits	N-Me	Ar-Go	Ar-Me	Go-Me
Frontal sinus																							
breadth	−0.171	−0.086	−0.106	−0.151	0.169	−0.066	0.132	0.117	−0.052	−0.008	0.125	−0.176	0.104	−0.045	0.12	−0.087	0.11	0.023	0.047	0.058	0.069	−0.042	−0.137
width	−0.169	−0.182	0.038	−0.247	0.224	−0.224	−0.148	0.023	0.048	−0.051	0.063	0.005	−0.023	−0.025	−0.053	−0.079	0.051	0.071	0.217	0.033	0.102	−0.139	−0.163
height	−0.038	0.012	−0.064	−0.130	0.154	−0.051	0.028	0.059	0.066	0.077	−0.046	0.024	−0.073	−0.025	0.05	−0.055	−0.046	−0.063	0.049	0.024	0.166	0.062	−0.079
volume	−0.107	−0.033	−0.095	−0.159	0.169	−0.062	0.093	0.082	−0.036	−0.005	0.108	−0.112	0.06	0.022	0.095	−0.059	0.015	−0.047	0.054	0.064	0.137	0.052	−0.090
**(b) Participants with skeletal Class I jaw–base relationship**																							
Pretreatment measurements																							
	Angular measurements													Linear measurements									
	SNA	SNB	ANB	Facial	Y-axis	Occ.pl.	Go.A	FMA	Pal pl.	U1-SN	IIA	IMPA	FMIA	SN	U1-NA	L1-NB	OJ	OB	Wits	N-Me	Ar-Go	Ar-Me	Go-Me
Frontal sinus																							
breadth	−0.171	−0.166	−0.212	−0.059	−0.157	−0.348	−0.156	−0.399	0.023	−0.058	0.06	0.263	0.067	0.163	−0.012	−0.189	0.032	−0.003	0.333	−0.272	0.269	−0.208	−0.241
width	−0.189	−0.165	−0.123	−0.111	−0.034	−0.348	−0.396	−0.410	−0.132	−0.060	0.101	0.378	−0.065	0.214	0.061	−0.032	0.024	−0.106	0.342	−0.117	0.346	−0.233	−0.192
height	−0.080	−0.036	−0.036	0.068	−0.220	−0.317	−0.073	−0.274	0.06	0.136	0.041	0.012	0.255	−0.028	0.133	−0.183	0.212	0.045	0.298	−0.335	0.02	−0.130	−0.041
volume	−0.203	−0.167	−0.167	−0.141	−0.124	−0.401	−0.251	−0.448 *	0.145	−0.026	0.037	0.342	0.016	0.207	0.044	−0.138	0.058	0.088	0.381	−0.387	0.269	−0.257	−0.333
Posttreatment measurements																							
	Angular measurements													Linear measurements									
	SNA	SNB	ANB	Facial	Y-axis	Occ.pl.	Go.A	FMA	Pal pl.	U1-SN	IIA	IMPA	FMIA	SN	U1-NA	L1-NB	OJ	OB	Wits	N-Me	Ar-Go	Ar-Me	Go-Me
Frontal sinus																							
breadth	−0.143	−0.117	−0.175	−0.369	0.212	−0.306	−0.236	−0.127	0.198	0.289	−0.327	0.365	−0.377	0.191	0.352	0.316	−0.043	−0.182	0.288	−0.179	0.191	−0.232	−0.263
width	−0.088	−0.280	−0.293	−0.315	0.184	−0.336	−0.417	−0.233	0.094	0.543	−0.367	0.312	−0.198	0.123	0.399	0.142	0.165	−0.112	0.332	−0.199	0.195	−0.153	−0.050
height	0.028	0.058	−0.114	−0.034	−0.073	−0.248	−0.084	−0.215	0.181	0.39	−0.479	0.487 *	−0.456	−0.023	0.153	0.158	−0.176	0.154	0.266	−0.321	0.123	−0.011	−0.077
volume	−0.046	−0.002	−0.200	−0.302	0.131	−0.280	−0.283	−0.236	0.164	0.367	−0.357	0.44	−0.371	0.165	0.38	0.333	−0.045	−0.063	0.275	−0.278	0.236	−0.175	−0.243
**(c) Participants with skeletal Class II jaw–base relationship**																							
Pretreatment measurements																							
	Angular measurements													Linear measurements									
	SNA	SNB	ANB	Facial	Y-axis	Occ.pl.	Go.A	FMA	Pal pl.	U1-SN	IIA	IMPA	FMIA	SN	U1-NA	L1-NB	OJ	OB	Wits	N-Me	Ar-Go	Ar-Me	Go-Me
Frontal sinus																							
breadth	0.088	−0.011	0.296	−0.173	0.142	0.258	0.166	0.123	0.079	−0.108	0.047	−0.006	−0.117	−0.253	−0.231	−0.026	−0.116	−0.108	−0.137	−0.158	0.019	−0.325	−0.312
width	−0.234	−0.150	−0.208	0.183	−0.133	0.027	−0.171	−0.241	−0.101	−0.108	0.006	0.266	−0.019	−0.175	0.016	0.061	−0.237	−0.014	0.172	−0.046	0.251	0.038	−0.092
height	−0.390	0.027	−0.121	−0.092	0.184	0.226	−0.085	0.071	−0.333	0.074	0.079	−0.183	0.182	−0.062	−0.077	−0.255	0.033	−0.503 *	−0.217	0.173	0.338	0.196	0.021
volume	−0.073	−0.086	0.057	−0.080	0.065	0.239	0.027	0.03	−0.124	−0.065	−0.021	0.111	−0.138	−0.073	−0.002	0.065	−0.067	−0.145	−0.129	−0.017	0.11	−0.074	−0.169
Posttreatment measurements																							
	Angular measurements													Linear measurements									
	SNA	SNB	ANB	Facial	Y-axis	Occ.pl.	Go.A	FMA	Pal pl.	U1-SN	IIA	IMPA	FMIA	SN	U1-NA	L1-NB	OJ	OB	Wits	N-Me	Ar-Go	Ar-Me	Go-Me
Frontal sinus																							
breadth	0.068	0.015	0.11	0.085	−0.035	−0.019	0.191	0.041	−0.556 *	−0.173	0.368	−0.356	0.458	−0.302	−0.364	−0.357	0.165	−0.058	0.045	−0.104	0.092	−0.301	−0.418
width	−0.187	0	−0.336	0.351	−0.264	0.051	−0.166	−0.294	−0.300	−0.097	0.312	−0.190	0.569 **	−0.225	−0.175	−0.569 **	−0.155	−0.079	−0.064	−0.004	0.319	0.043	−0.147
height	−0.390	0.027	−0.121	−0.092	0.184	0.226	−0.085	0.071	−0.333	0.074	0.079	−0.183	0.182	−0.062	−0.077	−0.255	0.033	−0.503 *	−0.217	0.173	0.338	0.127	−0.102
volume	−0.041	−0.010	−0.064	0.052	0.005	0.107	0.056	0.025	−0.573 *	−0.134	0.284	−0.266	0.347	−0.076	−0.234	−0.304	0.026	−0.257	−0.077	0.081	0.259	0.031	−0.158
**(d) Participants with skeletal Class III jaw–base relationship**																							
Pretreatment measurements																							
	Angular measurements													Linear measurements									
	SNA	SNB	ANB	Facial	Y-axis	Occ.pl.	Go.A	FMA	Pal pl.	U1-SN	IIA	IMPA	FMIA	SN	U1-NA	L1-NB	OJ	OB	Wits	N-Me	Ar-Go	Ar-Me	Go-Me
Frontal sinus																							
breadth	−0.387	−0.344	−0.282	−0.326	0.505	0.226	0.537	0.648 *	0.177	−0.584	0.476	−0.603 *	0.189	−0.020	−0.024	0.055	−0.288	−0.183	−0.295	0.459	−0.265	0.106	0.188
width	−0.445	−0.460	−0.064	−0.670 *	0.774 *	0.136	0.583 *	0.745 **	0.52	−0.560	0.436	−0.562 *	0.031	0.064	−0.021	−0.090	−0.001	−0.204	−0.161	0.329	−0.266	−0.277	−0.233
height	−0.118	−0.143	0.079	−0.391	0.499	0.06	0.552 *	0.512	0.278	−0.473	0.461	−0.492	0.171	−0.097	−0.355	−0.090	−0.304	0.101	−0.071	0.123	−0.181	−0.133	−0.154
volume	−0.207	−0.167	−0.221	−0.341	0.524	0.242	0.680 *	0.616 *	0.24	−0.524	0.492	−0.598 *	0.215	−0.033	−0.186	−0.035	−0.351	−0.040	−0.284	0.325	−0.188	0.056	0.004
Posttreatment measurements																							
	Angular measurements													Linear measurements									
	SNA	SNB	ANB	Facial	Y-axis	Occ.pl.	Go.A	FMA	Pal pl.	U1-SN	IIA	IMPA	FMIA	SN	U1-NA	L1-NB	OJ	OB	Wits	N-Me	Ar-Go	Ar-Me	Go-Me
Frontal sinus																							
breadth	−0.392	−0.362	−0.142	−0.369	0.584 *	−0.437	0.51	0.683 **	0.026	−0.378	0.323	−0.573 *	0.109	−0.041	0.218	0.025	0.177	0.306	0.249	0.495	−0.237	0.142	0.129
width	−0.486	−0.556	0.235	−0.649 *	0.790 **	−0.342	0.345	0.760 **	0.322	−0.517	0.241	−0.353	−0.152	−0.032	−0.018	−0.036	0.358	0.45	0.309	0.35	−0.290	−0.182	−0.203
height	−0.157	−0.181	0.085	−0.453	0.614 *	−0.483	0.401	0.639 *	0.265	−0.382	0.312	−0.426	−0.003	−0.045	0.051	0.042	0.083	0.247	0.38	0.272	−0.167	0.006	−0.098
volume	−0.227	−0.205	−0.096	−0.370	0.561 *	−0.399	0.603 *	0.668 *	0.122	−0.480	0.428	−0.580	0.126	−0.001	0.068	−0.041	0.032	0.241	0.248	0.421	−0.106	0.214	0.038

* *p* < 0.05; ** *p* < 0.01.

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
