# Peer review of "Volumetric Assessment of the Frontal Sinus in Female Adolescents and Its Relationship with Craniofacial Morphology and Orthodontic Treatment: A Pilot Study"

_ijerph, 2022, doi:10.3390/ijerph19127287_

Round 1

Reviewer 1 Report

Dear Authors, 

the topic of the airways measurements during the orthodontic and surgical procedures is widely discussed in the literature lately. The presented paper is generally nicely prepared, but needs several improvements:

1. In line 68, please specify what was the sample size

2. Please, modify table 1, so that it is clearer (either add divisions between the texts or bold the explained terms), also add the information which cephalometric analysis do the Authors use.

3. Due to the fact, that the research is investigated lately, the novel papers should be incorporated (at least 10 of the past 5 years) to the discussion, eg.

- Dastan F, Ghaffari H, Hamidi Shishvan H, Zareiyan M, Akhlaghian M, Shahab S. Correlation between the upper airway volume and the hyoid bone position, palatal depth, nasal septum deviation, and concha bullosa in different types of malocclusion: A retrospective cone-beam computed tomography study. Dent Med Probl. 2021;58(4):509–514. doi:10.17219/dmp/130099

DedeoÄŸlu N, Duman SB. Clinical significance of maxillary sinus hypoplasia in dentistry: A CBCT study. Dent Med Probl. 2020 Apr-Jun;57(2):149-156. doi: 10.17219/dmp/114982. PMID: 32602270.

Li Q, Tang H, Liu X, Luo Q, Jiang Z, Martin D, Guo J. Comparison of dimensions and volume of upper airway before and after mini-implant assisted rapid maxillary expansion. Angle Orthod. 2020 May 1;90(3):432-441. doi: 10.2319/080919-522.1.

Liu P, Jiao D, Wang X, Liu J, Martin D, Guo J. Changes in maxillary width and upper airway spaces in young adults after surgically assisted rapid palatal expansion with surgically facilitated orthodontic therapy. Oral Surg Oral Med Oral Pathol Oral Radiol. 2019 May;127(5):381-386. doi: 10.1016/j.oooo.2018.11.005. 

Inchingolo, A.D.; Ceci, S.; Patano, A.; Inchingolo, A.M.; Montenegro, V.; Di Pede, C.; Malcangi, G.; Marinelli, G.; Coloccia, G.; Garibaldi, M.; Kruti, Z.; Palmieri, G.; De Leonardis, N.; Rapone, B.; Mancini, A.; Semjonova, A.; Nucci, L.; Bordea, I.R.; Scarano, A.; Lorusso, F.; Ferrara, E.; Farronato, M.; Tartaglia, G.M.; Di Venere, D.; Cardarelli, F.; Inchingolo, F.; Dipalma, G. Elastodontic Therapy of Hyperdivergent Class II Patients Using AMCOP® Devices: A Retrospective Study. Appl. Sci. 202212, 3259. https://doi.org/10.3390/app12073259

Nota, A.; Caruso, S.; Caruso, S.; Sciarra, F.M.; Marino, A.; Daher, S.; Pittari, L.; Gatto, R.; Tecco, S. Rapid Maxillary Expansion in Pediatric Patients with Sleep-Disordered Breathing: Cephalometric Variations in Upper Airway’s Dimension. Appl. Sci. 202212, 2469. https://doi.org/10.3390/app12052469

Thank you!

Author Response

Reviewer #1:

The topic of the airways measurements during the orthodontic and surgical procedures is widely discussed in the literature lately. The presented paper is generally nicely prepared, but needs several improvements:

(Response)

Thank you for the nice wordings.

  1. In line 68, please specify what was the sample size

(Response)

The meaning of the sample size is the necessary sample size to meet the desired statistical constraints. (revision: Lines 68-69)

  1. Please, modify table 1, so that it is clearer (either add divisions between the texts or bold the explained terms), also add the information which cephalometric analysis do the Authors use.

(Response)

According to the reviewer’s suggestion, we added divisions (3 points) between the texts and changed the explained terms in bold. (revision: Lines 108-145) We used the cephalometric measurements by Downs, Northwestern, Tweed, and Jacobson method.

  1. Due to the fact, that the research is investigated lately, the novel papers should be incorporated (at least 10 of the past 5 years) to the discussion, eg.

(Response)

The reviewer is correct. Then, we added the following sentence at the beginning of the Discussion section and cited some references according to the reviewer’s request. (revision: Lines 358-359; References #12-16)

With the technological advancement in CT, growing evidence suggests the critical role of paranasal sinuses in craniofacial growth and orthodontic treatment.

Reviewer 2 Report

We read with great interest the manuscript with title: “Volumetric Assessment of the Frontal Sinus in Female Adolescents and Its Relation with the Craniofacial Morphology” aiming to assess the correlation between frontal sinus morphology and craniofacial morphology, and to investigate the effects of orthodontic treatment on the frontal sinus development in female adolescents.

The manuscript is of interest; however, several criticisms need to be addressed before resubmission.

1.     Please change the title, in my opinion the most interesting part of the present work is the correlation of the changes in sinus dimensions with the orthodontic treatment.

I would like therefore to suggest a new title: “Volumetric Assessment of the Frontal Sinus in Female Adolescents and Its Relation with Orthodontic Treatment and Craniofacial Morphology. A pilot study”

2.     The greatest criticism of this study is the lack of a control group, i.e., subjects who did not undergo orthodontic treatment. for this reason, I propose to insert the words " A pilot study " in the title and I ask you to underline this limitation with a paragraph titled "Limitations" at the end of the discussion.

3.     In addition, in the Materials and methods section is required to identify orthodontic therapies that were carried out in subgroups and to explain why the study population is only female, while in fact meeting the WHO criteria, it should consist of equal parts of both sexes.

Author Response

Reviewer #2:

We read with great interest the manuscript with title: “Volumetric Assessment of the Frontal Sinus in Female Adolescents and Its Relation with the Craniofacial Morphology” aiming to assess the correlation between frontal sinus morphology and craniofacial morphology, and to investigate the effects of orthodontic treatment on the frontal sinus development in female adolescents.

The manuscript is of interest; however, several criticisms need to be addressed before resubmission.

(Response)

Thank you for the nice wordings.

  1. Please change the title, in my opinion the most interesting part of the present work is the correlation of the changes in sinus dimensions with the orthodontic treatment.

I would like therefore to suggest a new title: “Volumetric Assessment of the Frontal Sinus in Female Adolescents and Its Relation with Orthodontic Treatment and Craniofacial Morphology. A pilot study”

(Response)

Thank you for the excellent suggestion. According to the reviewer’s suggestion, we changed the title into “Volumetric Assessment of the Frontal Sinus in Female Adolescents and its Relationship with Craniofacial Morphology and Orthodontic Treatment: A Pilot Study”

  1. The greatest criticism of this study is the lack of a control group, i.e., subjects who did not undergo orthodontic treatment. for this reason, I propose to insert the words " A pilot study " in the title and I ask you to underline this limitation with a paragraph titled "Limitations" at the end of the discussion.

 (Response)

The reviewer is correct. According to the reviewer’s suggestion, one more paragraph about the limitations was added at the end of the Discussion section (revision: 440-445):

This study had some limitations. First, the study did not include any control samples, that is, participants who did not undergo orthodontic treatment. Our results showed small, but significant, increases in the size and volume of the frontal sinus during orthodontic treatment; however, without the control data, we could not distinguish whether these increases were affected by orthodontic treatment. It is difficult to record 3D CT images for individuals who are not undergoing treatment because of ethical concerns.

  1. In addition, in the Materials and methods section is required to identify orthodontic therapies that were carried out in subgroups and to explain why the study population is only female, while in fact meeting the WHO criteria, it should consist of equal parts of both sexes.

    (Response)

We totally agree with the reviewer’s opinion. We have considerable number (20-30) of CT data taken from male adolescents before and after orthodontic treatment; however, the sample size was still smaller and not sufficient to meet the desired statistical constraint. If we had a sufficient number of the male samples, we were supposed to use all data from both male and female adolescents. In addition, previous studies did not report gender-difference in the size and volume of the frontal sinus (Goldman-Yassen et al., 2021; Said et al., 2017). On the other, we agree that further study using both the male and female adolescents is needed to examine a gender-related difference in the dimensions of frontal sinus. We added the following sentences as the second limitation of this study (revision: Lines 445-451):

Second, our participants were only females, and we could not estimate the sex-dependent differences in the dimensions of the frontal sinus. Furthermore, the effect of orthodontic treatment on the sinus dimensions may be different between male and female adolescents. Previous studies also did not report a sex-based-difference in the size and volume of the frontal sinus; therefore, further studies involving both male and female participants are needed.

Round 2

Reviewer 1 Report

Thank you for the correction done.

Reviewer 2 Report

Dear Authors,

Thank you for having improved the manuscript.

For new future and more detailed data, I invite you to publish a fuller report on this subject.

At this time, after your revisions, the manuscript text is ready to move on to the next stages of publication.

Best regards

Author Response

Reviewer 2

Thank you for having improved the manuscript.

For new future and more detailed data, I invite you to publish a fuller report on this subject.

At this time, after your revisions, the manuscript text is ready to move on to the next stages of publication.

(Response)

Thank you for the nice wording. According to your request, we will submit the other manuscript when we complete a fuller report on this subject in the future.